# A substrate-multiplexed platform for profiling enzymatic potential of plant family 1 glycosyltransferases

Sasilada Sirirungruang [1,2,3,4,11], Vincent Blay [2,11], Elys P. Rodriguez[5,6], Yasmine F. Scott[2,7], Khanh M. Vuu [2], Collin R. Barnum[8,9], Paul H. Opgenorth[2], Fanzhou Kong[5,6], Yuanyue Li [6], Oliver Fiehn [5,6] & Patrick M. Shih [1,2,3,10] ✉

Plants have expanded various biosynthetic enzyme families to produce a wide diversity of natural products; however, most enzymes encoded in plant genomes remain uncharacterized, highlighting the need for new functional genomic approaches. Here, we report a platform enabling the rapid functional characterization of plant family 1 glycosyltransferases, which serve important roles in plant development, defense, and communication. Using substrate-multiplexed reactions, mass spectrometry, and automated analysis, we screen 85 enzymes against a diverse library of 453 natural products, for a total of nearly 40,000 possible reactions. The resulting dataset reveals a widespread promiscuity and a strong preference for planar, hydroxylated aromatic substrates among family 1 glycosyltransferases. We also characterize glycosyltransferases with an unusually wide substrate scope and with a non-canonical Cys-Asp catalytic dyad. This work establishes a widely-applicable enzymatic screening pipeline, reflects the immense glycosylation capability of plants, and has implications in biocatalysis, metabolic engineering, and gene discovery.

The explosion of sequence information in the post-genomic era promises insights into cells and organisms. While genome and transcriptome information have offered unique opportunities to understand the fundamentals of cellular inner workings, functional discovery of the encoded proteins remains a major challenge and the rate-limiting step in understanding an organism's metabolic capabilities[1,2]. Traditionally, biosynthetic gene discovery efforts focus on target pathways and characterize enzymes one by one, lacking behind the pace of sequencing. As a result, there is a significant gap between sequence information generation and interpretation. Hence, approaches to perform systematic functional characterization of enzymes at the genome scale can be a way to facilitate functional determination.

Biosynthetic gene discovery in plants is especially challenging because of the high complexity of plant genomes and the genetic intractability of most plant species[3,4]. Notably, plants contain an extensive repertoire of chemicals fundamental to their development, communication, and defense. A key class of biosynthetic enzymes integral to many facets of plant metabolism and physiology are glycosyltransferases (GTs)[5,6]. Nearly all small molecule glycosylation in plants is catalyzed by family 1 GTs, which also constitute the largest enzyme family found in plant genomes[7]. Small molecule glycosylation

[1]Department of Plant and Microbial Biology, University of California, Berkeley, CA, USA. [2]Joint BioEnergy Institute, Emeryville, CA, USA. [3]Environmental Genomics and Systems Biology Division, Lawrence Berkeley National Laboratory, Berkeley, CA, USA. [4]Center for Biomolecular Structure, Function and Application, Suranaree University of Technology, Nakhon Ratchasima, Thailand. [5]Department of Chemistry, University of California, Davis, CA, USA. [6]West Coast Metabolomics Center, University of California, Davis, CA, USA. [7]Department of Molecular and Cell Biology, University of California, Berkeley, CA, USA. [8]Department of Plant Biology, University of California, Davis, CA, USA. [9]Biochemistry, Molecular, Cellular, and Developmental Biology Graduate Group, University of California, Davis, CA, USA. [10]Innovative Genomics Institute, University of California, Berkeley, CA, USA. [11]These authors contributed equally: Sasilada Sirirungruang, Vincent Blay. ✉e-mail: pmshih@berkeley.edu

*in planta* serves various functions, including molecular storage, transport, detoxification, and sequestration[8,9]. Moreover, many glycoside natural products are important therapeutics, highlighting their relevance to human health[10,11]. Given the importance of plant glycoside natural products, a major challenge has been identifying the substrate selectivity of GTs. Although there have been efforts to characterize family 1 GTs[12–14], only a small fraction of all family 1 GTs across all sequenced plant genomes and transcriptomes have been experimentally characterized, creating a disparity between sequence availability and functional understanding.

Developing a platform to functionally characterize plant GTs *en masse* has the potential to accelerate our capacity to elucidate and expand metabolism. High-throughput enzymology studies will allow us to more broadly understand the basic structure-function relationships of whole enzyme families, informing protein engineering efforts and improving functional predictions. Mass spectrometry (MS)-based approaches are well suited to characterize family 1 GTs, as the addition of a sugar moiety onto an acceptor substrate leads to a consistent mass shift according to the identity of the sugar donor. In addition, MS can identify individual glycoside products from a complex pool of metabolites, making it amenable to multiplexing many substrate candidates in a single reaction mixture, which dramatically increases the assay

throughput compared to single-substrate reactions. Thus, MS is a suitable method to assess glycosylation reactions in a large-scale, multiplexed manner.

In this work, we develop a multiplexed MS-based platform to rapidly and systematically assay GT function on a large set of small molecules. A total of 85 family 1 GTs from *Arabidopsis thaliana* are subcloned from a synthetic library[15] into an *Escherichia coli* expression vector. We then use the *E. coli* lysate as the enzyme source to screen a diverse natural product library of 453 compounds in batches against all enzymes combinatorially. Our findings provide a unique genome-scale, protein-family-wide perspective on the enzymatic range and chemical diversity of glycosides that can be produced by Arabidopsis, helping to define its genome-wide biosynthetic glycosylation capacity.

## Results

### A substrate-multiplexed platform to elucidate the function of plant family 1 GTs

We set out to develop an enzymatic assay to assess sugar acceptor selectivity of family 1 GT activities on a genome-wide scale. We first cloned 85 Arabidopsis family 1 GTs from a previously generated library[15] into an *E. coli* expression vector pET28a (Fig. 1a, Supplementary Data 1). Family 1 GTs from Arabidopsis have been classified into 15

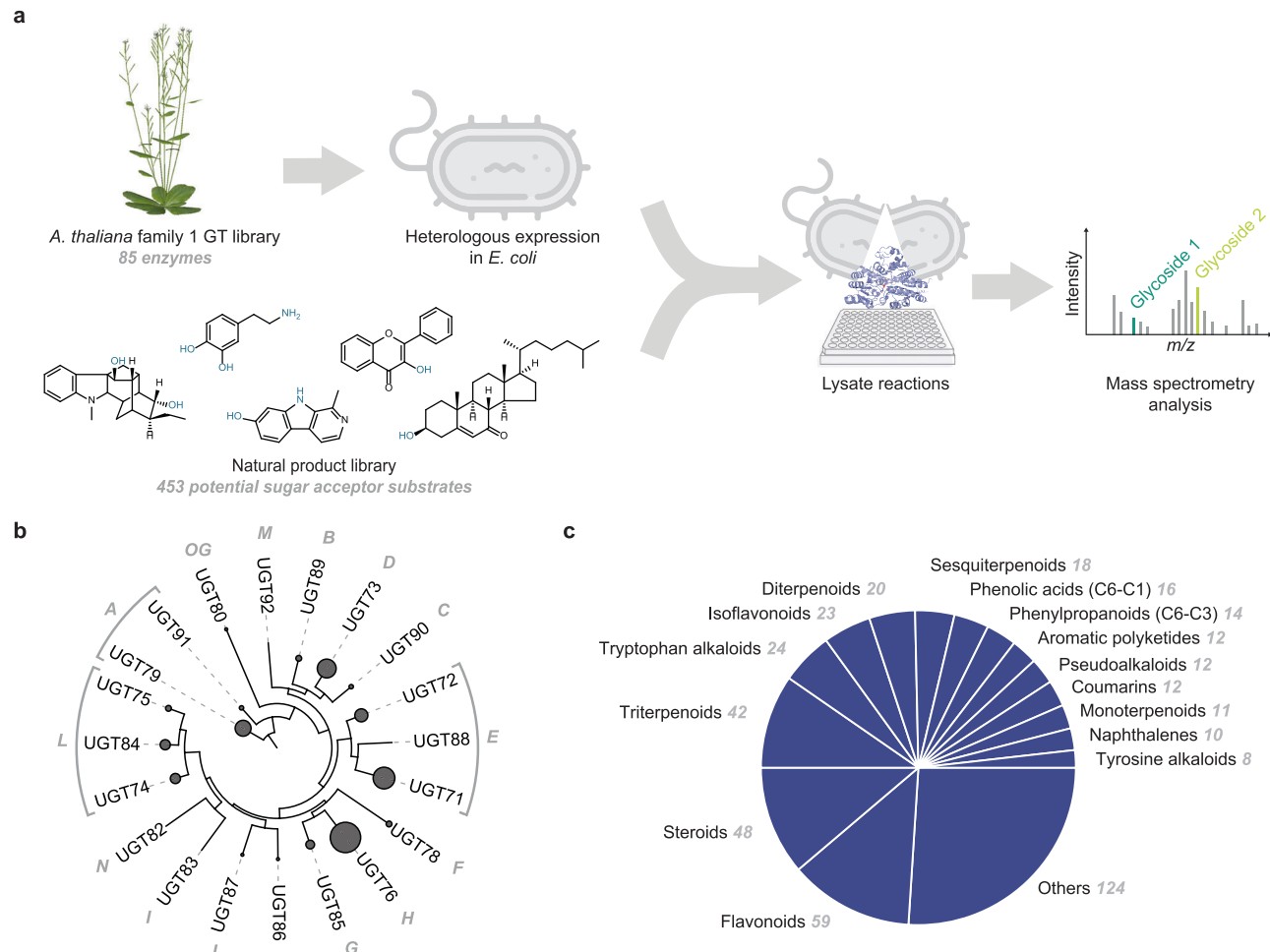

**Fig. 1 | Development of a substrate-multiplex functional screen for enzymatic glycosylation activities. a** Family 1 GT enzymes from Arabidopsis were expressed in *E. coli*, whose lysate was then used as the enzyme source to glycosylate 453 sugar acceptor substrate candidates selected from a natural product library. Lysate-based reaction mixtures were analyzed by high-resolution MS for glycosylation products without replicate (*n* = 1). **b** Arabidopsis family 1 GT enzymes are classified

into 15 phylogenetic clades indicated in gray capital letters, each containing 1–3 families indicated with UGT family number. The size of the gray circle represents the number of enzymes in each family found in the Arabidopsis genome. **c** Sugar acceptor substrate candidate library spans 42 natural product superclasses, 14 of which were represented by at least ten molecules. The number of molecules in each superclass is shown in gray italics.

phylogenetically distinct clades, 14 of which have the canonical GT-B fold and are referred to as groups A through N[16] (Fig. 1b). To expedite the experiments and circumvent the challenges associated with protein purification, we used cell lysates as the source of enzymatic activity. We first conducted pilot studies that confirmed the feasibility of this approach: the glycosylation activity of clarified lysate of *E. coli* expressing UGT75B1 was compared against that of purified enzyme and found to have comparable activity on known substrates, measured by liquid chromatography with tandem mass spectrometry (LC-MS/MS) (Supplementary Fig. 1). This finding was further confirmed on ten other family 1 GT enzymes under the same reaction conditions, demonstrating our ability to scale this approach (Supplementary Fig. 1). Next, we sought to multiplex sugar acceptor substrates in each reaction to increase the throughput of the platform.

Family 1 GTs have been previously reported to act on many types of substrates[12,17]; therefore, to comprehensively capture their potential activity, we aimed to assay each individual enzyme against a diverse set of potential sugar acceptors. We selected 453 compounds from Analyticon Discovery's MEGx natural product library (www.ac-discovery.com) based on the presence of nucleophilic functional groups, namely hydroxyl, amine, thiol, or aromatic ring, as they are required for glycosylation (Supplementary Data 2). The resulting acceptor substrate library spanned over 42 compound superclasses, 14 of which were represented by at least ten molecules (Fig. 1c). Library members were pooled into sets of 40 molecules with unique molecular weights to enable the detection of distinct compounds by MS (Fig. 1a). Each enzyme was assayed with a set of 40 substrates in a single reaction in order to balance the throughput and the sensitivity of our platform.

Family 1 GTs from Arabidopsis were expressed in *E. coli* similarly to previously reported[18], screened against all 453 potential sugar acceptors in multiplexed batches of 40, and analyzed using LC-MS/MS. A total of 38,505 reactions were thus screened. UDP-glucose was selected as the sole sugar donor in the screen because of its high abundance in cells[19], reported wide acceptance by plant family 1 GTs[12,20], and low cost compared to other nucleotide sugar donors. Lysate of *E. coli* expressing GFP was used as a negative control. Following the incubation of clarified *E. coli* lysate expressing individual GTs with UDP-glucose sugar donor and 40 sugar acceptor candidates overnight, the crude reaction mixture was dried, resuspended in methanol, and injected into LC-MS/MS for analysis. Data-dependent acquisition was performed with inclusion lists containing all possible single- and double-glycosylation products from each reaction. The screen resulted in 1,020 chromatograms.

## A computational analysis pipeline for the identification of glycosides in mass spectra of complex mixtures

To facilitate the analysis of the large number of mass spectra acquired, we developed a computational pipeline to systematically identify glycosides from a complex mixture. Mass features were first extracted from mass chromatograms for their precursor *m/z* and associated MS/MS spectra. Afterwards, they were analyzed based on the assumptions that a mass feature of an enzymatic glycosylation product would: 1) have its precursor exact mass equal that of the aglycone substrate +162.0533 for single-glycosylation and +324.1066 for double-glycosylation product, and 2) display an MS/MS fragmentation pattern similar to that of its corresponding aglycone substrate[21–23] (Supplementary Fig. 2). Extracted mass features were compared to a library of reference spectra curated from MassBank of North America (MoNA; massbank.us). The similarity between an experimental MS/MS spectrum and a reference spectrum was calculated using the cosine score, which measures the vectorial alignment between the relative intensities of matching fragment ions across both spectra[24].

The cosine score ranges from 0 to 1, with higher values indicating more similar spectra.

A stringent cosine score threshold of 0.85 was used to indicate positive enzymatic glycosylation products. This stringent threshold was chosen based on literature precedence to minimize the false discovery rate across various quality of reference and experimental spectra[25,26]. Using the cutoff of 0.85, glycosides of seven substrate candidates, namely 7,8-hydroxyflavone, biochanin A, hesperetin, naringenin, phloretin, (*S*)-isocorydine, and tryptamine, were observed in the lysate of *E. coli* expressing GFP negative control and were dropped from analysis. With this criterion, the automated pipeline identified a total of 4230 putative reaction products, including 3,669 single glycosides and 561 double glycosides (Supplementary Data 3). As expected, the cosine score threshold affects the number of reactions determined as productive (Supplementary Fig. 3; Results using cosine score thresholds of 0.75, 0.8, and 0.9 are provided in Supplementary Data 4, 5, and 6, respectively). As such, we performed further investigation to ensure that the chosen threshold is appropriate for this study.

To validate the lysate screen and the automated analysis pipeline, the results were compared to those of another large-scale activity screening of family 1 GT enzymes[12]. Yang et al. previously reported the activities of 54 of family 1 GT from Arabidopsis against 91 substrates. There are 582 overlapping reactions with our study, which involve 17 substrates and 36 enzymes. Of those reactions, there was agreement on ~70% of their outcomes using the cosine score threshold of 0.85, despite the studies being conducted under different experimental conditions (Supplementary Fig. 4). While Yang et al. performed glycosylation reactions at pH 7.8 with 177 μM of UDP-glucose and 0.1 mg/mL of substrates, our study performed reactions at pH 6.8 with 83 μM of UDP-glucose and 10 μM of substrates.

In addition, our lysate screening results were validated through in vitro reactions with purified enzymes without multiplexing. To confirm the lysate-based screening results, ten selected substrates were obtained, and ten selected GTs were cloned, expressed as GST fusions, and purified (Supplementary Fig. 5). Substrates were chosen to represent a wide variety of compounds, both in terms of structure and activity. The ten substrates range from those expected to be accepted by few to over a hundred enzymes and are structurally diverse. They span multiple superclasses: terpenes, steroids, flavonoids, isoflavonoids, coumarins, fatty acids, amino acids, and alkaloids. Reactions without enzymes were included as negative controls. Out of the 100 reactions investigated, 75 reactions resulted in outcomes in agreement with the lysate screening result (Supplementary Fig. 6), yielding 75% accuracy. Precision, recall, specificity, and F1 (mean of precision and recall) values were 77%, 65%, 83%, and 71%, respectively.

We looked into the incorrect calls of reaction outcomes by the automated pipeline. On the one hand, reactions that were productive only in purified enzyme reactions (false negatives, i.e., UGT71C4, UGT72B2, and UGT75D1 on osajin) displayed weak signals that resulted in no or low-quality MS/MS spectra in lysate-based screen. On the other hand, reactions that seemed productive only in the lysate screen (false positives, i.e., UGT71C4 and UGT72B2 on aleuritic acid; UGT75B2 and UGT87A2 on cortisone) did show signals of the expected product masses in both reaction conditions, but the signals differed in retention times. In addition to these edge cases, we analyzed the relationship between productive reactions and various aspects of experimental and reference spectra to determine possible sources of systematic error. We found that the number of reference spectra entries and the information content of those spectra did not affect the likelihood of finding glycosylation products (Supplementary Figs. 7 and 8). Moreover, no bias was observed in the number of reference spectra or the average number of peaks in those spectra among substrate supergroups (Supplementary Fig. 7 and 8). The only

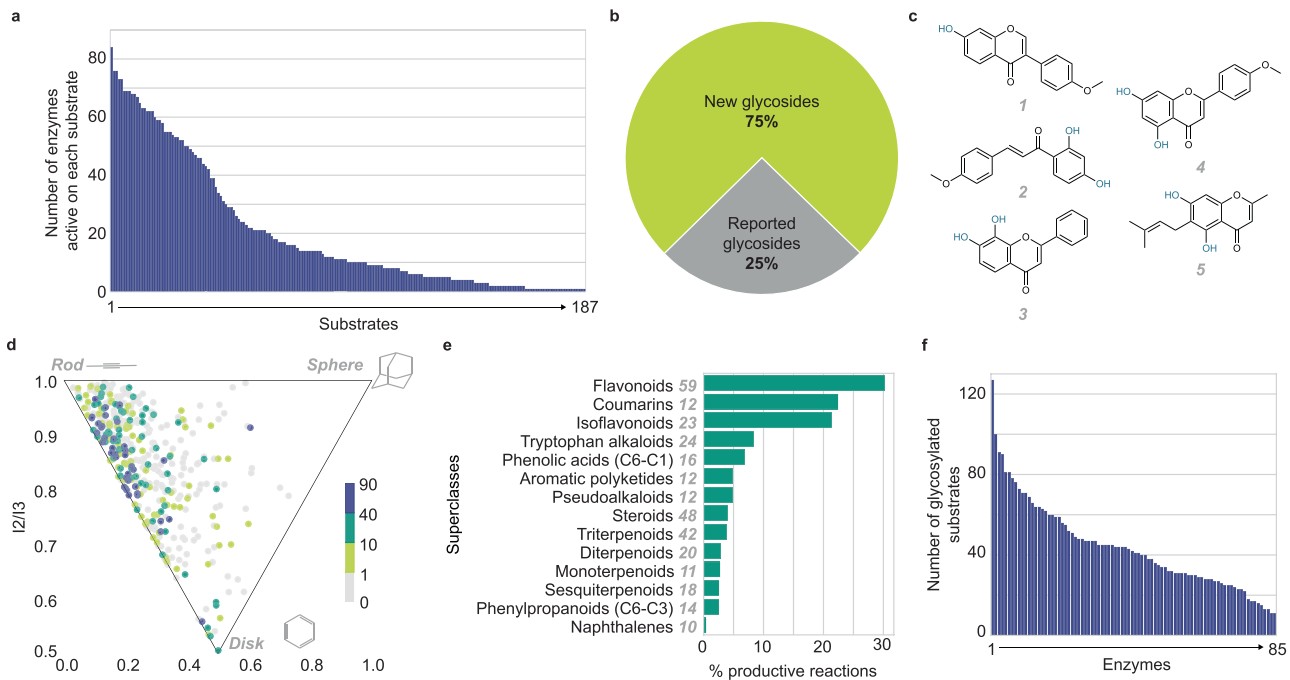

**Fig. 2 | Overview of large-scale, lysate-based screening results. a** Sugar acceptor substrates are ordered according to the number of family 1 GTs that were found to glycosylate them in the lysate screen. The screen found that 187 out of 453 substrate candidates were glycosylated by at least one enzyme, while the most reactive substrates are accepted by most enzymes. **b** The majority of glycosylation products identified from the lysate-based screen are not found in the PubChem database. **c** The most reactive substrates (1: formononetin, 2: 2′, 4′-dihydroxy-4-methoxychalcone, 3: 7,2′-dihydroxyflavone, 4: acacetin, 5: peucenin) share common structural features: they consist of hydroxylated aromatic rings, assume planar configuration, and are similar in surface area and volume. Nucleophilic functional groups in the molecules are shown in green. **d** Principal moments of inertia analysis shows that substrates (colored dots) in this study tend to be more rod-like and disk-like than non-substrates (gray dots). Substrates are colored based on the number of enzymes found to glycosylate them in the lysate screen as follows (color, number of enzymes): purple, > 40 enzymes; green, 10–40 enzymes; yellow, 1–10 enzymes. **e** The percentages of glycosylation reactions (enzyme + substrate combinations) that were productive in the lysate screen are shown by substrate superclasses. Flavonoids, coumarins, and isoflavonoids are most likely to be glycosylated, with over 30.2%, 22.5%, and 22.4% of reactions being productive, respectively. The number of molecules in each superclass is shown in gray italics. **f** Family 1 GT enzymes in this study are ordered by the number of substrates they glycosylated in the lysate screen. All 85 enzymes were active on at least 11 substrates. The median enzyme glycosylated 41 substrates, and the most active enzyme glycosylated 127 substrates. Source data are provided as a Source Data file.

factor that we found affecting the likelihood of finding glycosylation products through our lysate screen coupled with automated data analysis pipeline seems to be the product signal intensity (Supplementary Fig. 9).

Overall, from both the comparison to a prior study and to follow up in vitro reactions, we have demonstrated a high level of reproducibility when utilizing a cosine score threshold of 0.85. It is not surprising that the lysate-based screen does not reflect every enzymatic activity with complete accuracy as our platform was designed to balance accuracy and speed; nevertheless, the breadth of our dataset serves as an empirically derived starting point for enzyme selection of individual glycosylation reactions of interest.

### Sugar acceptor preference and enzyme promiscuity revealed by large-scale family 1 GT characterization

Our dataset revealed that a large and diverse variety of molecules can serve as the sugar acceptor substrates in enzymatic glycosylation. We identified 187 sugar acceptor substrates (41.3% of the library) that could potentially be glycosylated by at least one enzyme (Fig. 2a). Putative substrates spread across 34 superclasses and include both plant and non-plant metabolites (Supplementary Data 3, Supplementary Fig. 10 and 11). The wide sugar acceptor scope reflects family 1 GTs' function in expanding the chemical diversity of plants and their various roles in development, speciation, and defense. Notably, 75% of glycosylation products observed in this study were not found in the

PubChem database (Fig. 2b), highlighting the potential of these enzymes to unlock new natural products.

Global analysis of our dataset reveals clear biases and trends in the preferences of sugar acceptors by family 1 GTs. In fact, our data suggests that while most sugar acceptor candidates were not substrates, the most popular substrates (*i.e.*, formononetin; 2′,4′-dihydroxy-4-methoxychalcone; 7,2′-dihydroxyflavone; acacetin, and peucenin) were glycosylated by over 70 enzymes in our family 1 GT library. These molecules share similar structural features, including hydroxylated aromatic rings and planar configurations (Fig. 2c). This is consistent with the comparison of normalized principal moments of inertia ratios for substrates and non-substrates, which revealed that family 1 GTs prefer molecules that are rod- or disk-like, with low sphericity (Fig. 2d). Additionally, we found that family 1 GT exhibits a bias towards smaller and less globular molecules without regard to hydrophobicity (Supplementary Fig. 12).

Our dataset also reveals clear substrate superclass preferences across family 1 GT enzymes. Of all superclasses of molecules in this study, flavonoids, coumarins, and isoflavonoids are most likely to be glycosylated, with over 30.2%, 22.5%, and 22.4% of enzyme-substrate reaction pairs being productive, respectively (Fig. 2e, Supplementary Fig. 10 and 11). These results are in line with the countless glycoside derivatives of the three superclasses of molecules observed in nature. For example, over 300 kaempferol-*O*-glycosides and 300 quercetin-*O*-glycosides have been reported[27]. Given the variable, and widespread nature of flavonoids, coumarins, and isoflavonoids in plants, this

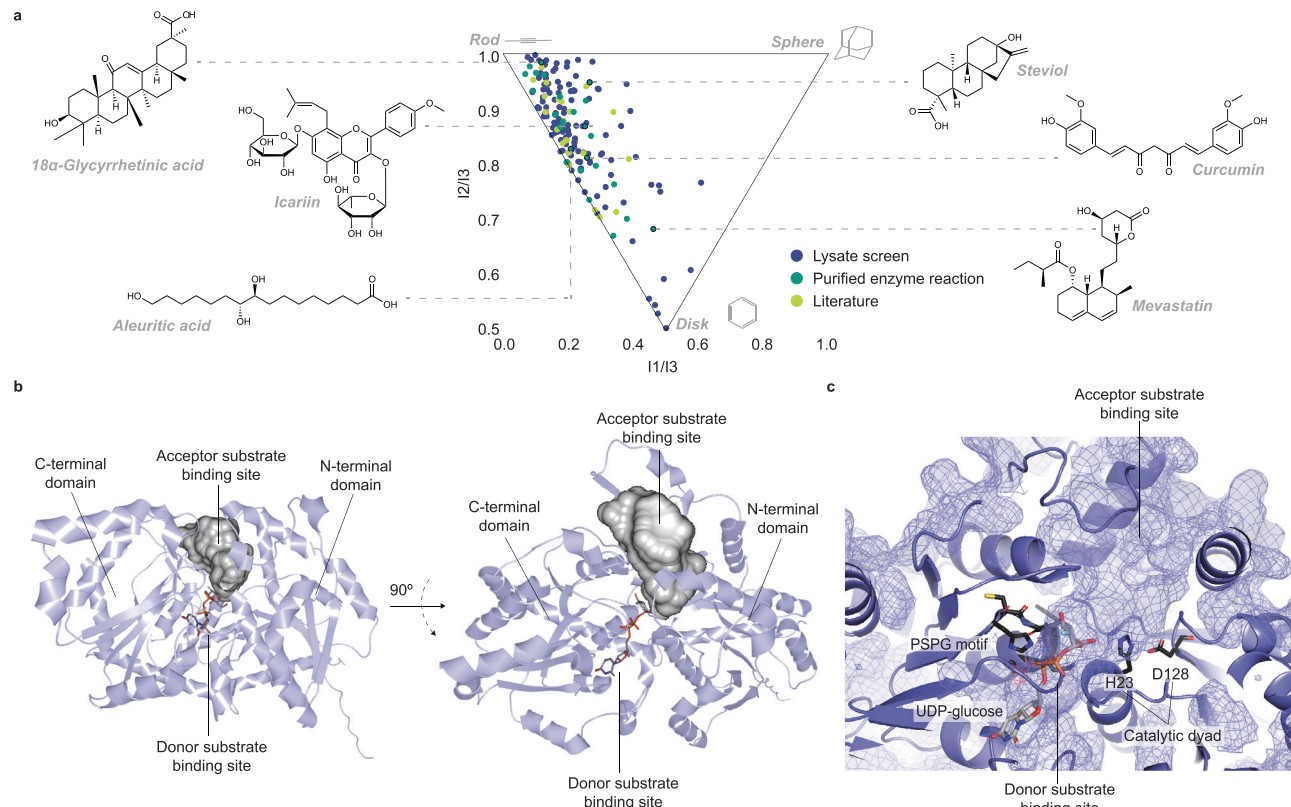

**Fig. 3 | Characterization of the highly promiscuous family 1 GT UGT73C5.**
**a** Principal moments of inertia analysis shows that sugar acceptor substrates of UGT73C5 are structurally diverse. 25 of 127 substrates found in the lysate screen were confirmed in purified enzyme reactions, including 18α-glycyrrhetinic acid (top left), icariin (middle left), aleuritic acid (bottom left), steviol (top right), curcumin (middle right), and mevastatin (bottom right). Substrates that were found glycosylated in the lysate screen (n = 1) are shown in purple, and substrates that were validated by purified enzyme reactions (n = 2–3) are shown in green. Substrates reported in the literature are shown in yellow. **b** Investigation of the structural model of UGT73C5 (blue ribbons) reveals an expansive 1600 Å³ acceptor substrate binding site (gray surfaces) next to the donor substrate binding site (gray sticks) between the N-terminal and the C-terminal domains of the enzyme. **c** The large sugar acceptor binding pocket and the sugar donor binding site are found between the catalytic dyad (H23-D128; black sticks, right) and the central part of the PSPG motif (HCGWNS; black sticks, left) of the enzyme. The protein surface is shown in a semi-transparent blue surface and mesh.

observation may reflect plants' needs for numerous distinct ways to glycosylate them. This observation may also reflect a greater challenge to achieve regioselectivity in hydroxylated polycyclic aromatic systems, which have planar configurations with few rotatable bonds and strong nucleophilicity, making them excellent GT substrates.

By contrast, terpenoids are much less likely to be glycosylated by family 1 GTs, with under 5% of the enzyme-substrate reaction pairs studied being productive for each class of terpenoids (Fig. 2e). This narrower scope can be due to terpenoids being more variable in size, shape, and position of their nucleophilic handles than flavonoids, coumarins, and isoflavonoids. This variability makes it less likely that any one enzyme active site can accommodate many different terpenoids. Other small hydrophilic molecules such as monosaccharides and amino acids are also not observed glycosylated in the lysate screen. However, these superclasses are underrepresented in our substrate library and may require further investigation.

From the enzyme perspective, our dataset reveals a widespread sugar acceptor substrate promiscuity and a high degree of redundancy in biochemical capacity among family 1 GTs. Every enzyme included in this study was found to potentially accept at least 11 substrates. The median number of sugar acceptor substrates accepted was 41, and the most promiscuous enzyme could glycosylate up to 127 substrates (Fig. 2f). Note that an enzyme's productivity in our study results from a combination of expression level, kinetics, stability in vitro, and sugar acceptor substrate promiscuity. Thus, enzymes that appear less productive may suffer from low expression, slow kinetics, low stability, or

inability to utilize UDP-glucose as a sugar donor. In addition, practically every enzyme glycosylates some flavonoid molecules. In fact, four out of the five most widely accepted substrates are flavonoids. This observation showcases a striking degree of biochemical redundancy among family 1 GTs in the genome. Overall, our findings expand on previous reports on the sugar acceptor substrate promiscuity and selectivity of family 1 GTs[17,20,28].

**Unprecedented substrate promiscuity of family 1 GT revealed by multiplexed screening**

The systematic and agnostic nature of our substrate-multiplexed platform led to notable discoveries regarding family 1 GTs. The large size and diversity of our substrate library revealed unprecedented sugar acceptor promiscuity among family 1 GTs. The most promiscuous enzyme in our study, UGT73C5, accepts an impressively broad range of sugar acceptors. UGT73C5 was found to glycosylate up to 127 substrates from 29 compound superclasses in the lysate-based screen, in comparison to previous studies where in vitro activity for UGT73C5 was reported on eight compound superclasses[12,29–32]. Twenty five of the 127 putative substrates from 17 superclasses were validated by purified enzyme reactions (Supplementary Fig. 13-17). Validated substrates are functionally and structurally diverse and include both plant and non-plant metabolites (Fig. 3a). In addition, UGT73C5 has previously been reported to have in vitro activity on 15 other molecules[12,29–32] and *in planta* activity on brassinosteroid hormones[29] and deoxynivalenol[33] (Fig. 3a). Even though our dataset does not allow

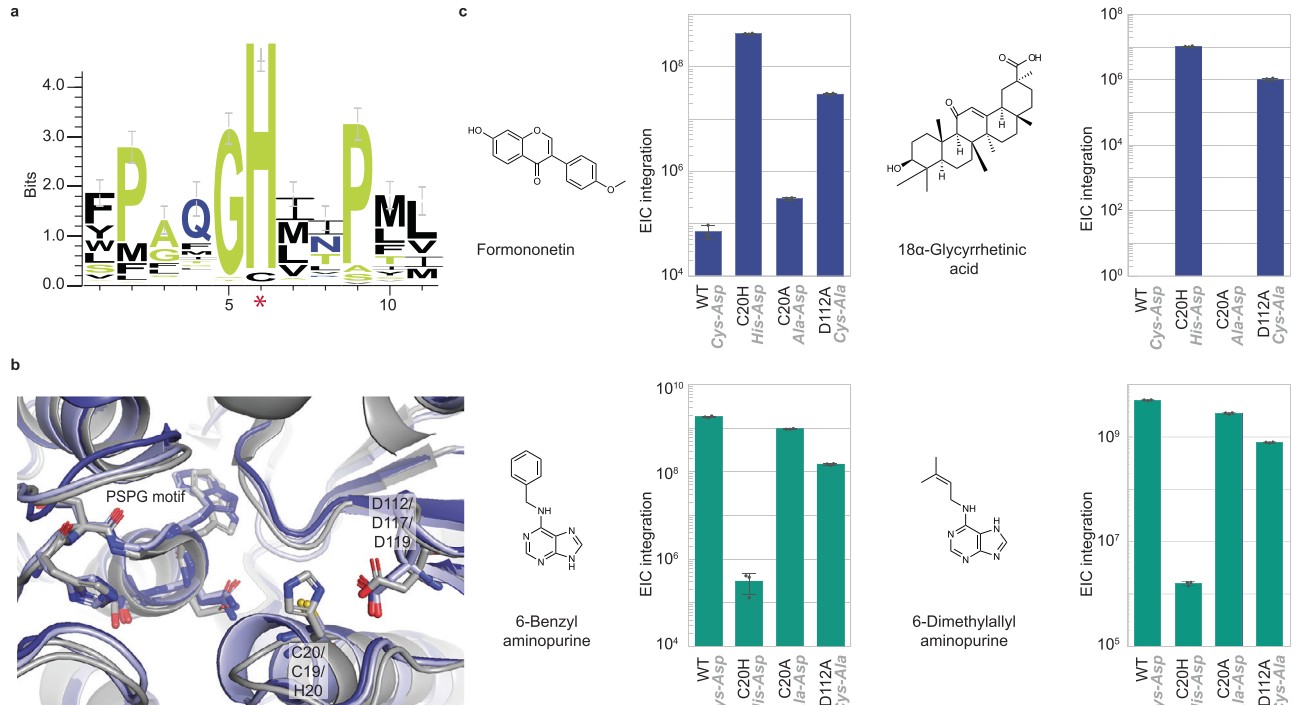

**Fig. 4 | Characterization of *O*- and *N*-glycosylation activities of family 1 GT enzymes with non-canonical Cys-Asp catalytic dyad. a** Sequence logo of the catalytic (*) and its surrounding residues of all Arabidopsis family 1 GTs shows the highly conserved His is replaced with Cys in a small subset of enzymes, namely UGT76C1-5. **b** The superimposition of the active site architecture of structural models of UGT76C2 (dark blue), UGT76C3 (light blue), and VvGT1 crystal structure(gray; PDB ID# 2C1Z) shows that C20 of UGT76C2 and C19 of UGT76C3 assumes the same position as catalytic H20 of VvGT1. Catalytic dyads and the central part of the PSPG motif (HCGWNS) are shown in sticks and labeled with residue numbers of UGT76C2/UGT76C3/VvGT1. **c** Integrations of extracted ion chromatograms of glycosylation products of formononetin (top left), 18α-glycyrrhetinic acid (top right), 6-benzylaminopurine (bottom left), and 6-dimethylallylaminopurine (bottom right) produced by UGT76C2 wild-type and catalytic dyad single mutants show that *O*-glycosylation reactions (top, blue) are favored by UGT76C2 C20H and UGT76C2 D112A while *N*-glycosylation reactions (bottom, green) are favored by UGT76C2 WT and UGT76C2 C20A. The catalytic dyad in each enzyme variant is shown in gray italics. Structures of sugar acceptor substrates are shown to the left of bar plots. Data are mean ± s.d. of three technical replicates (*n* = 3). Source data are provided as a Source Data file.

for the direct quantitative comparison of catalytic efficiencies on different substrates, it comprises the largest and most structurally diverse set of substrates accepted by a single plant family 1 GT enzyme to the best of our knowledge.

To understand the basis of the extreme promiscuity of UGT73C5, we investigated its structural model from the AlphaFold Protein Structure Database[34]. The inspection revealed an expansive and easily accessible putative sugar acceptor binding pocket next to the catalytic dyad, the plant secondary product glycosylation (PSPG) motif[35], and the sugar donor binding site (Fig. 3b, c). With a pocket of roughly 10 ✕ 20 Å opening and nearly 25 Å in depth, it is conceivable that the 1600 Å³ cavity can house a great diversity of chemical structures. Furthermore, the internal walls of the cavity comprises mostly hydrophobic residues on all sides (F18, A20, I88, F101, F129, V195, P196, V197, P202, W206, F210, F395, A396, P420, and W423); thus, it is possible that the putative binding pocket can adjust to accommodate organic molecules of various sizes and shapes. In addition, we compared the size of the putative binding pocket of UGT73C5 to those of other most and least promiscuous enzymes according to the lysate-based screen. We observed that the four most promiscuous enzymes tend to have larger cavity volumes (enzyme, number of glycosylated substrates, cavity volume in Å³: UGT73C5, 127, 1600; UGT71D1, 100, 1825; UGT73B3, 91, 746; UGT73B4, 90, 1733) than the four least promiscuous enzymes (enzyme, number of glycosylated substrates, cavity volume in Å³: UGT89B1, 11, 669; UGT76E12, 11, 951; UGT71B6, 13, 1149; UGT85A3, 13, 301) (Supplementary Fig. 18 and 19). This finding is consistent with previous reports that drew a positive correlation between the size of substrate binding site and enzyme promiscuity[36,37].

However, there likely are additional factors that contribute to the promiscuity levels observed in our study. For example, although UGT85A3 may not be able to accept many substrates because of its small cavity size of 301 Å³, UGT71B6 has a large cavity of 1,149 Å³ and glycosylated just as few substrates in the lysate screen. Indeed, we only observed a weak positive correlation between the number of glycosylated substrates and putative sugar acceptor substrate binding size when the analysis was expanded to the entire dataset of 85 enzymes (Supplementary Fig. 20). Furthermore, we did not observe any correlation between the number of glycosylated substrates and enzyme isoelectric point or ESM-2 pseudo-perplexity, which are proxies for solubility and stability[38] (Supplementary Fig. 20). As such, our findings suggest that substrate promiscuity of family 1 GTs is a complex trait that may be governed by multiple factors, including the size and the hydrophobicity of binding pockets.

### *N*-glycosylation selectivity of bifunctional family 1 GTs with non-canonical Cys-Asp catalytic dyad

Data from our lysate-based screen suggests that the canonical catalytic dyad His-Asp may not be required for *O*-glycosylation activity. Four enzymes in our study, UGT76C1-3 and 5, have cysteine in place of histidine in the catalytic dyad (Fig. 4a). Investigation of their structural models generated by AlphaFold2 showed that the cysteine residue is found in the same position as that typical of the catalytic histidine in other family 1 GT enzymes (Fig. 4b). The *O*-glycosylation activities of UGT76C2 and UGT76C3 observed in the lysate-based screen were validated using purified enzymes, formononetin, and UDP-glucose. Weak signals were observed corresponding to formononetin

glucoside with exact mass, fragmentation pattern, and retention time matching those of the product generated by other enzymes with canonical catalytic dyad in this study (Supplementary Fig. 21). This finding contrasts with a previous report arguing that catalytic histidine is required for *O*-glycosylation activity[39]. This distinction is likely due to the differences in enzymes, substrates, reaction conditions, and detection method sensitivity.

The non-canonical Cys-Asp catalytic dyad displays a strong preference for *N*-glycosylation over *O*-glycosylation. UGT76C2 has been shown to be responsible for *N*-glycosylation of cytokinins in Arabidopsis[40,41]. We confirmed the finding using in vitro purified enzyme reactions and showed that UGT76C2 and UGT76C3 enzymes yielded strong mass signals corresponding to the glycosylation products of 6-benzylaminopurine and 6-dimethylallylaminopurine (Supplementary Fig. 22). Thus, we hypothesized that the non-canonical Cys-Asp catalytic dyad confers UGT76C2 and UGT76C3 a higher selectivity for *N*-glycosylation. To test this hypothesis, we mutated the active site cysteine residue (C20 in UGT76C2 and C19 in UGT76C3) to histidine and tested the variants' *O*-glycosylation (formononetin and 18α-glycyrrhetinic acid) and *N*-glycosylation (6-benzylaminopurine and 6-dimethylaminopurine) activities. Indeed, we observed that UGT76C2 and UGT76C3 generated more *N*-glycosylation products than their corresponding histidine mutants, whereas UGT76C2 C20H and UGT76C3 C19H generated more *O*-glycosylation products under the same reaction conditions than their wild-type counterparts (Fig. 4c, Supplementary Fig. 23). Equivalently, the ratios of *N*- to *O*-glycosylated products obtained with wild type are higher than those obtained with histidine mutant enzymes. As such, our data suggests that these enzymes have adopted the non-canonical catalytic dyad Cys-Asp to enable *N*-glycosylation selectivity.

The precise roles of cysteine and aspartate active site residues in catalysis were further investigated by mutagenesis and docking. We mutated each residue individually to alanine, creating UGT76C2 C20A, UGT76C2 D112A, UGT76C3 C19A, and UGT76C3 D117A. All variants were tested for *O*- and *N*-glycosylation activities. Both residues appear to contribute to catalysis, as mutating either to alanine reduced *N*-glycosylation and increased *O*-glycosylation (Fig. 4c, Supplementary Fig. 23) compared to corresponding wild-type enzymes. We speculate that while the canonical His−Asp catalytic dyad facilitates *O*-glycosylation by acting as a catalytic base in an $S_N2$-like mechanism, UGT76C2 and UGT76C3 may facilitate *N*-glycosylation simply by bringing the two substrates within proximity of each other at the correct orientation, as previously suggested for trifunctional *O*-/*N*-/*S*-glycosyltransferase *Pt*UGT1[39] and bifunctional *O*-/*N*-glycosyltransferase UGT72B1[42]. A preliminary docking experiment supports our hypothesis (Supplementary Fig. 24). Alternatively, the non-canonical Cys-Asp catalytic dyad may act through another mechanism. The cysteine residue in Cys-Asp dyad may form S-aromatic interactions with the purine acceptor substrates to optimally orient them for *N*-glycosylation[43,44], act as a conjugate base in an $S_N1$-like mechanism[45,46], or serve as a nucleophile in a double displacement-like mechanism[47,48]. The precise mechanism driving the observed chemoselectivity will require further investigations.

## Discussion

In this work, we show that activities of family 1 GTs can be assessed *en masse*. We successfully developed a substrate-multiplexed enzymatic assay that is broadly applicable to family 1 GTs and applied it to Arabidopsis enzymes at the genome scale. We carried out combinatorial screens of 85 enzymes against 453 potential sugar acceptor substrates and UDP-glucose for a total of 38,505 reactions, greatly expanding the reaction space explored for family 1 GTs. Multiplexing substrates allowed us to achieve scale by streamlining efforts in both experimental setups and analysis, without requiring liquid-handling robots. Moreover, we developed a fast, automated computational pipeline to analyze mass chromatograms and identify glycosylation products in complex mixtures of *E. coli* lysate and small molecules. Adapted from an approach widely used among metabolomic studies, our pipeline facilitated large biochemical mass spectrometric data collection and drastically accelerated the typically tedious, manual curation step. Our process resulted in a total of 4230 putative enzyme activities, many of which are likely previously unreported transformations. In fact, ~75% of all glycoside products identified in our study were not previously reported in the largest public chemical database PubChem. This suggests that family 1 GTs may provide access to an expanded repertoire of glycosides available for fine chemical and pharmaceutical discovery purposes.

In addition, our study revealed insights into the family 1 GT enzymes themselves. Overall, we observed that many family 1 GTs are highly promiscuous in terms of sugar acceptors, especially on disk-like and rod-like molecules such as flavonoids, coumarins, and isoflavonoids. We found a GT enzyme with an unprecedented substrate scope that may provide a unique scaffold for future enzyme engineering efforts. We also characterized GTs that used a non-canonical Cys-Asp catalytic dyad, rather than the classical His-Asp dyad. Although our results do not speak to enzyme functions in vivo, our genome-wide functional characterization of the whole family 1 GTs revealed unforeseen metabolic capacity of the organism and provides a wealth of information invaluable for biocatalysis and metabolic engineering applications.

As an attempt to catalog putative enzyme reactions at an unprecedented scale, several limitations to our design and analysis should be noted. First, the reported platform does not take into account the potentially variable expression levels of different family 1 GT enzymes in *E. coli*. Thus, family 1 GTs with low expression levels may appear less active in the lysate-based screening result. Second, our approach does not allow regioselectivity determination. Even though the number of product isomers produced by an enzyme can often be determined by the number of individual mass features that elute at different retention times, the products were not isolated for further characterization, which is necessary to determine the precise position of glycosylation. Furthermore, the experiment was performed with only UDP-glucose as the sugar donor. As such, GT enzymes that have strong selectivity for another sugar donor will appear less active in our results. In addition, some substrates may be inherently incompatible with our experimental setup. For example, monoterpenoid glycosylation was not detected in this study even though such activity has been reported among Arabidopsis GTs[49]. Lastly, this platform does not allow quantitative assessment of enzyme products. As MS sensitivity is different for every molecule, it is not possible to directly compare the efficiency of a particular enzyme across substrates using this platform without access to glycoside standards.

Although much attention was given to the design of the platform and the computational analysis pipeline, it is inevitable that the lysate-based screen does not reflect every enzymatic activity with complete accuracy. We expected there to be increased variability and noise inherent to scaling up a process and thus naturally a tradeoff between scale and accuracy. Indeed, some disagreements were found between the lysate-based screen and purified enzyme reactions, and between our results and a previously reported dataset[12]. The differences between experimental datasets may be a result of inhibition by other substrates or products in the reaction mixture, interfering activity of *E. coli* lysate, differing experimental conditions between studies, or the limitation of the automated product identification pipeline. All of these reflect the complexity associated with generating large, reliable biochemical datasets. As large-scale datasets of protein functions receive increasing attention thanks to advances in sequencing, DNA synthesis, and artificial intelligence, we hope that our study provides valuable lessons for future similar efforts.

We expect that our platform can be adapted and improved to apply to other enzyme families. Enzyme families in which substrates can be wide-ranging and the enzymatic activity result in distinct and predictable mass shifts in products (*e.g.*, transferases) are good candidates for substrate-multiplexed platforms. Substrate-screens for enzyme families have previously been reported, but none explored sequence and chemical spaces as extensively as reported here (This study: 38,505 reactions; GTs: 6,318 reactions[12] halogenases: 10,788 reactions[50] esterases: 13,920 reactions[36] and phosphatases: ca. 33,000 reactions[51]. Widespread application of this technology may facilitate the discovery of enzyme functions and advance our understanding of the metabolic capacity of organisms. It may also open doors to pathway engineering and expand the chemical space accessible through biosynthesis. Overall, our work sets an example for the design, execution, and analysis of an enzymatic platform of unprecedented scale that has led to a massive dataset regarding family 1 GT enzymatic function. Our study is broadly applicable and translatable to other enzyme families and has implications in functional genomics, biocatalysis, and metabolic engineering.

## Methods

### Bacterial strains
*E. coli* XL1 Blue was used for DNA construction. *E. coli* Rosetta(DE3) pLysS and BL21*(DE3)-T1$^R$ were used for heterologous protein expression.

### Gene and plasmid construction
Standard molecular biology techniques were used to carry out plasmid construction. All PCR amplifications were carried out with Phusion High Fidelity DNA polymerase with primer annealing temperatures 4–8 °C below the $T_m$. DNA assembly was performed using the isothermal Gibson assembly protocol. For analysis and isolation of large DNA fragments from agarose gel, 1 % gel and QIAquick PCR Purification Kit (QIAGEN) was used. All constructs were verified by sequencing (UC Berkeley DNA Sequencing Facility, Berkeley, CA; and Genewiz from Azenta Life Sciences, South Plainfield, NJ).

**pET28-AtGT1.** Coding sequences for family 1 GT enzymes were amplified from pDONR-GT library[15]. pET28a constructs were cloned by assembling PCR products into pET28a via Gibson assembly. *E. coli* cells transformed with the assembly mixture were selected on culture medium containing carbenicillin.

**pGEX-6P-1-AtGT1.** Coding sequences for family 1 GT enzymes were amplified from pDONR-GT library[15]. GST-UGT fusion constructs were cloned by assembling PCR products into the BamHI site of pGEX-6P-1 via Gibson assembly. *E. coli* cells transformed with the assembly mixture were selected on culture medium containing carbenicillin.

### Protein expression and lysis for lysate-based enzymatic screening platform
pET28-AtGT1 plasmids were transformed into *E. coli* Rosetta(DE3) pLysS, and cells were grown in 100 mL of LB medium in 250 mL baffled flasks at 37 °C with shaking at 200 rpm to $OD_{600}$ of ~0.75. Protein expression was induced with IPTG to final concentration of 0.15 mM, and cells were then grown at 20 °C for 18 h. Afterwards, 100 mL of cultures were harvested by centrifugation and resuspended in a lysis buffer (20 mM Tris, 100 mM NaCl, 5 mM imidazole, pH 6.8). Cell suspension was aliquoted to 1 ml aliquots, and lysed using a Fisher Scientific Series 60 Sonic Dismembrator Model F60 (5" L × 1 1/7" W resonant body that steps down to a 1/7" diameter probe) with two cycles of 10 s ON and 30 s OFF. Lysate was clarified by centrifugation at 14,000 g for 15 min at 4 °C, and the supernatant was immediately used as the source of family 1 GT.

### Sugar acceptor substrate selection and grouping
The library of potential sugar substrate candidates were assembled by selecting compounds from the Analyticon natural product library that contain at least one nucleophilic group that has been reported to participate in enzymatic glycosylation reactions, namely hydroxyl, amine, thiol, carboxylic acid, and aromatic groups. Compounds were divided into those retained on reverse phase columns and on normal phase columns and grouped into pools of 40, named mixes 1-12. All mixes contain compounds compatible with reverse phase liquid chromatography.

Superclasses of molecules were assigned based on their canonical SMILES by querying them against NPClassifier[52] (https://npclassifier.ucsd.edu). A few molecules that were assigned to more than one superclass by the software were reassigned to the most common superclass based on the substrate list. A few molecules that were not assigned to any superclass were denoted "NA".

PMI plots were generated based on the canonical SMILES of each molecule using rdkit 2023.09.4 (https://www.rdkit.org/). SMILES were used to initialize molecule objects, hydrogen atoms were added, and the molecules were embedded in 3D coordinate space with 10 conformers per molecule. Each of the conformers were then optimized using the MMFF94 forcefield[53] in rdkit. For each molecule, the conformation with the lowest energy was used to compute the principal moments of inertia (PM1, PM2, PM3) using rdkit Descriptors3D module. In the figures, we show the relative moments of inertia PM1/PM3 and PM2/PM3.

### Lysate-based glycosylation reaction
Substrate mixes were prepared by combining 120 μL of 10 mM solution in DMSO of each of the 40 substrates to the final volume of 4.8 mL, resulting in the final concentration of 0.25 mM for each substrate. Reaction mixtures (1 mL) contained 229.5 μL lysis buffer, 16.65 μL of 5 mM UDP-glucose, 40 μL substrate mix, 20 μL of 3 M NaCl, 32.7 μL of 500 mM MgCl$_2$, and 666 μL cell lysate. All reactions were performed in 2-mL, 96-well blocks. Glycosylation reactions were incubated at 30 °C overnight with shaking at 120 rpm. Following incubation, reactions were dried by using CentriVap Concentrator (LabConCo), and crude residue was resuspended in 100 μL of methanol for analysis.

### LC-MS/MS analysis of lysate-based glycosylation reactions
All LC-MS/MS analysis was carried out on the Thermo Fisher Scientific Vanquish UHPLC system connected to the Q Exactive Orbitrap Mass Spectrometer. The Q Exactive was operated using positive mode electrospray ionization with the following parameters: mass range, 120–2000 *m/z*; sheath gas flow rate, 60; aux gas flow rate, 25; sweep gas flow rate, 2; spray voltage (kV), 3.6; capillary temp, 300 °C; S-lens RF level, 50; aux gas heater temp, 370 °C. Full MS parameters are: resolution, 70,000; AGC target, 1e6; maximum IT, 100 ms; spectrum data type, centroid. Data-dependent MS2 parameters are: inclusion, on; resolution, 17,500; AGC target, 4e3; maximum IT, 100 ms; loop count, 4; top N, 4; isolation window, 1.0 *m/z*; fixed first mass, 70.0 *m/z*; (N)CE 65; spectrum data type, centroid. An inclusion list with the expected *m/z* values of the glycosylation products was created for each substrate mix.

For reverse phase analysis, 2 μL of samples were separated on a Waters Acquity UHPLC BEH column (2.1 × 100 mm; 1.7 μm particle size) at 40 °C at the flow rate of 0.4 mL/min without splitting. The mobile phases consisted of (A) water (100%) with formic acid (0.1%) and (B) acetonitrile (100%) and formic acid (0.1%). The separation was conducted under the following gradient: 0 min 1% (B); 0–1 min 1% (B); 1–15 min 99% (B); 15–18 min 100% (B); 18 min 1% (B); 18–20 min 1% (B). Data were collected using the Thermo Fisher Scientific XCalibur 4.3 software and analyzed using the Thermo Fisher Scientific Freestyle 1.6 software.

## LC-MS/MS spectral processing

Raw files were converted to .mzML centroid format using MS Convert[54]. Individual mass features and their associated MS/MS spectra were extracted using AlphaPept[55]. Only features with the precursor ion $m/z$ matching those expected of $[M+H]^+$, $[M+Na]^+$, $[M+NH_4]^+$ and $[M-H_2O+H]^+$ adducts of single and double glycosylation products within 3 mDa were considered. Subsequently, the associated MS/MS spectra were processed. Precursor ions and all fragments larger than $[M+H-1.6]^+$ were removed from MS/MS spectra. Fragment ions with an intensity less than 1% of the highest-abundance ion were also eliminated. Fragment ions were then binned to 20 mDa bins, and total fragment intensities were normalized among spectra.

## Automated glycoside identification

A single and double glycosylation product MS/MS spectral library was simulated from an equivalent aglycone MS/MS spectral library curated from the Mass Bank of North America (MoNA, https://mona.fiehnlab.ucdavis.edu, accession date: 26-May-2024). The precursor $m/z$ values of all aglycones in the substrate library were shifted by +162.0533 Da and +324.1066 Da to match the expected precursor $m/z$ values of mono- and di-glycosides. In the assignment of precursor peaks, the following adducts were then considered possible: $[M+H]^+$, $[M+Na]^+$, $[M+2H]^{2+}$, $[M+NH4]^+$, and $[M+ACN+H]^+$.

For each MS/MS fragmentation spectrum acquired, the precursor $m/z$ for each MS/MS signal was compared to the calculated $m/z$ of all possible glycosylation product adducts in that reaction mix. Any precursor within 50 ppm was considered a candidate. The MS/MS fragmentation spectrum was then compared to all reference fragmentation patterns for each candidate obtained from MoNA. A cosine similarity score was obtained for each pair of fragmentation patterns compared using the CosineGreedy score algorithm in matchms 0.25.0[56] with the following settings: tolerance=0.15 Da, mz_power=0, intensity_power=1.0. The intensities of the spectra being compared were each normalized by the corresponding strongest peak, and only up to the 50 strongest peaks were used in the comparison. The fragmentation spectrum was assigned to the candidate with the highest cosine score as long as it exceeded 0.5 (this storage cutoff was later increased as discussed in the text).

The area under the curve (AUC) for each glycosylation product was estimated by integrating the corresponding peak in the MS channel using a custom algorithm, which implements a simplified trapezoidal integration in the $m/z$ and time dimensions. This procedure was repeated for each MS/MS fragmentation spectrum recorded by the instrument, across all the reaction mixes analyzed. The pipeline was initially run on a set of enzyme-free cell lysates as controls. All glycosylation products identified by the pipeline in these controls were excluded from the results.

To estimate the number of glycosylation products that may be present in PubChem we used E-utilities, the public API to the NCBI Entrez system (https://eutils.ncbi.nlm.nih.gov/entrez/eutils/esearch.fcgi). Each compound in our substrate list was queried in E-utilities by name, using the term "{name} *glucoside". A glycosylation product was considered potentially present in Pubchem if the corresponding query returned at least one CID chemical identifier.

## Expression of GST-tagged family 1 GT enzymes for purification

Plasmids encoding the proteins of interest were transformed into *E. coli* BL21*(DE3)-T1$^R$. TB culture medium (0.5 L) with appropriate antibiotic (carbenicillin: 100 µg/mL) in a 2.5 L ultra-yield flask was inoculated with overnight culture of freshly transformed *E. coli* cells. Cells were grown at 37 °C with shaking at 200 rpm to $OD_{600} = 0.8$–1.2, at which time, they were cold-shocked on ice for 20-40 min. Expression was induced by addition of 0.5 mM of IPTG. Cells were then grown at 16 °C with shaking at 200 rpm overnight and harvested by

centrifugation at $10,000 \times g$ for 5 min at 4 °C. Cell pellets were stored at −80 °C until purification.

## Purification of GST-tagged family 1 GT enzyme

Cell pellets were resuspended in a lysis buffer (500 mM sodium phosphate, 150 mM sodium chloride, pH 7.5) containing lysozyme (0.5-1 mg/mL) at 5–8 mL/g of cell pellet. Cell suspensions were then sonicated using 5 sec ON/25 sec OFF cycle for a total ON time of 2 min. Cleared cell lysates were then obtained after centrifugation at $14,000 \times g$ for 20 min at 4 °C. Cleared cell lysates were incubated with 1–2 mL of Glutathione Sepharose 4B (Cytiva) for 1 h at 4 °C before loading onto the column by gravity flow. The column was washed with 5–15 mL of lysis buffer and with wash buffer (50 mM Tris, 150 mM sodium chloride, pH 8.0) until the eluate was negative for protein content when tested by Bradford protein assay reagent (Bio-Rad). Prescission protease was added to the column and incubated overnight. On the following day, protein was eluted from the column with Wash Buffer B. Eluted protein was concentrated using a 30 kDa MWCO Amicon Ultra spin concentrator (Millipore).

Protein concentration was determined using A280 and protein extinction coefficients calculated from the sequence using the ExPASY ProtParam program. For enzyme assays, purified protein was aliquoted and flash-frozen in liquid nitrogen before storing at −80 °C. Column was cleaned with 30 mL of Elution Buffer C (50 mM Tris, 10 mM glutathione, pH 8.0) and re-equilibrated in 30 mL of Lysis Buffer A before the next use. Yields of protein preparations are as follow (enzyme, yield in mg/L of culture): UGT71C4, 1.2; UGT72B1, 2.6; UGT72B2, 1.5; UGT73C4, 1.2; UGT73C5, 5.5; UGT74D1, 0.9; UGT75B2, 0.9; UGT75D1, 1.0; UGT76C2 and variants, 0.3-0.5; UGT76C3 and variants, 0.2-0.3; UGT76C5, 0.6; UGT87A2, 1.7.

## In vitro enzyme assay for UDP-dependent glycosylation

All assay mixtures (25 µL) contained 20 mM Tris, 150 mM sodium chloride, 20 mM magnesium chloride, 5 mM imidazole, 500 µM UDP-glucose, 50 µM substrate, and 10 µM enzyme at pH 7.6. Reactions were incubated at 25 °C for 1 h. Reactions were quenched with 2 µL of 1 M trichloroacetic acid, and 3 µL of 50 µM quercetin was added as an internal standard. The reactions were then extracted with 550 µL of ethyl acetate and vortexed for 15-30 sec. The organic layer (250 µL) was transferred and dried using CentriVap Concentrator (LabConCo). Samples were kept at −80 °C until analysis.

For reactions involving UGT76C2 and UGT76C3 and their variants, to allow for a semi-quantitative assessment of enzymatic activity, a modified protocol was applied. Assay mixtures containing the same components described previously were incubated at 25 °C for 2.5 h. Reactions were quenched with 75 µL of cold 50% methanol in water containing 40 mM of trichloroacetic acid and 120 µM of quercetin as an internal standard. Samples were kept at −80 °C until analysis.

## LC-MS analysis of glycosylation products in purified enzyme reactions

Samples were resuspended in 50 µL of 50% methanol in water and filtered through a 96-well 0.2 µM PVDF membrane plate by centrifugation at 1000 g for 5 min. Samples (5 µL for reactions involving UGT76C2 and UGT76C3 and their variants and 2 µL for all other reactions) were analyzed by Q Exactive Orbitrap mass spectrometer (Thermo Fisher Scientific) equipped with an electrospray ionization source in positive ionization mode. Samples were analyzed on a Waters Acquity UPLC BEH C18 column (2.1 × 100 mm, 1.7 µm particle size) connected to Vanquish UHPLC (Thermo Fisher Scientific) using a gradient from 1 to 99% of acetonitrile with 0.1% formic acid in water with 0.1% formic acid in 17 mins at flow rate of 0.4 mL/min following an initial hold at 1% acetonitrile with 0.1% formic acid for 1 min at 40 °C. Data were collected using Xcalibur data acquisition software (Thermo

Fisher Scientific) and analyzed using FreeStyle 1.8 software (Thermo Fisher Scientific).

## Approximation of acceptor substrate binding site volume
Structural models of family 1 GT enzymes were downloaded from AlphaFold Protein Structure Database[34,57] The binding coordinates of sugar donor was approximated by superimposing protein structural models to the doubly bound crystal structure of VvGT1 (PDB ID# 2C1Z) and transferring the coordinates of the bound UDP-2-deoxy-2-fluoroglucose (U2F) from the crystal structure to protein models[58]. The structural models with superimposed U2F were then used as inputs for cavity volume computation using Caver Analyst 2.0[59] using probe sizes of 1.8 and 4.0 Å.

## Calculation of enzyme physical characteristics
The isoelectric point of each protein was estimated using the ProteinAnalysis module in BioPython 1.83[60] based on the primary protein sequence. The pseudo-perplexity (PPL) of each protein sequence was calculated based on the ESM2 protein language mode[61] I implemented in HuggingFace (checkpoint "facebook/esm2_t33_650M_UR50D"). For a given protein, each of the residues was sequentially masked when input to the model, and the average model loss for each sequence was then computed. This averaged loss was exponentiated to obtain the pseudo-perplexity.

## Docking
Multi-ligand molecular docking was conducted on selected protein structural models using the software AutoDock Vina 1.2.0[62,63]. PDBQT files were prepared using the software OpenBabel 2.4.1 and considering a pH of 7.4. The search space was defined by manually specifying a 3D rectangular box of $-4 \leq x \leq 11.5$, $-9.5 \leq y \leq 11.5$, and $-5 \leq z \leq 9$. The default grid spacing of 0.375 Å was used. A very high exhaustiveness setting of 128 was used to increase the number of searches in the optimization. The top scoring poses were saved and manually inspected using PyMOL 2.5.5.

## Reporting summary
Further information on research design is available in the Nature Portfolio Reporting Summary linked to this article.

## Data availability
Raw mass spectrometry data files are available at Zenodo at record number 15670423. All nucleotide sequences used in this manuscript are available at Joint BioEnergy Institute's Inventory of Composable Elements (ICE) (https://public-registry.jbei.org). Source data are provided with this paper.

## Code availability
All custom codes used in this study are available at Github [https://github.com/vinbl/gtscreen/tree/main].

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

## Acknowledgements

We thank Gabrille Wyatt, Jay Yung, and the Zerbe lab for assistance with mass spectrometry. We thank Dr. Andy Zhou for assistance with gathering structural models of all enzymes in the study. We thank Dr. Edward Koleski for helpful discussions on enzyme mechanisms. This work was part of the DOE Early Career Award and the DOE Joint BioEnergy Institute (http://www.jbei.org) supported by the U.S. Department of Energy, Office of Science, Office of Biological and Environmental Research through contract DE-AC02-05CH11231 (P.M.S.) between Lawrence Berkeley National Laboratory and the U.S. Department of Energy. The United States Government retains and the publisher, by accepting the article for publication, acknowledges that the United States Government retains a non-exclusive, paid-up, irrevocable, worldwide license to publish or reproduce the published form of this manuscript, or allow others to do so, for United States Government purposes. E.P.R. was supported by NIH grant 1 T32 GM 136597-1 A1. P.M.S. and equipment were supported by grant number R00AT009573 from the National Center for Complementary and Integrative Health (NCCIH) at the National Institutes of Health.

## Author contributions

S.S. curated lysate-based screen data, cloned pGEX-6P-1-AtGT1 plasmids, expressed and purified proteins, designed, executed, and analyzed purified enzyme reactions, computed enzyme active site volumes, and wrote the manuscript. V.B. curated lysate-based screen data, curated reference spectrum library, designed and executed computational analysis pipeline of LC-MS data, computed substrate and enzyme physical characteristics, and wrote the manuscript. E.P.R. designed and executed the lysate-based screen. Y.F.S. cloned pGEX-6P-1-AtGT1 plasmids, and expressed and purified proteins. K.M.V. cloned pET28-AtGT1 plasmids. C.R.B. assisted with collecting LC-MS data. P.H.O. assisted with the design and optimization of the lysate-based screen. F.K. and Y.L. provided helpful discussions. O.F. provided substrate candidate library and financial support. P.M.S. conceptualized the study, provided financial support and supervision, and wrote the manuscript.

## Competing interests

P.M.S. has financial interest in BasidioBio. The other authors declare no competing interests.
