## [Peer Review file · Nature Communications]

A substrate-multiplexed platform for profiling enzymatic potential of plant family 1 glycosyltransferases

Corresponding Author: Dr Patrick Shih

Version 0:

Reviewer comments:

Reviewer #1

(Remarks to the Author)

I appreciate the authors comments and the modifications made to the manuscript and I accept the manuscript as is. 'Novelty' is a subjective term and I admire the effort and outcomes of the screening effort to determine glycosyltransferases promiscuity. And overcoming technical difficulties is to be praised, but that doesn't make the individual techniques novel. Also, I don't know what the authors mean the number of "events". In a single chromatogram you can have >10,000 features and with peak finding algorithms you can find >1,000 of peaks from those features. Those peaks need than to be scored in order to determine the number of actual identified compounds. The term "events" is not frequently used in metabolomics.

Reviewer #2

(Remarks to the Author)

The authors' responses to reviewers are generally comprehensive and targeted, addressing different review comments with in-depth replies and improving the research content through supplementary experimental data and revised descriptions. After revisions, the overall quality of the manuscript meets publication requirements. It makes significant contributions to the field of glycosyltransferase research in terms of methodological approaches and data accumulation. Through systematic screening and multi-dimensional validation, the study reveals the substrate promiscuity and catalytic mechanisms of plant family 1 glycosyltransferases, providing rich empirical data and new research perspectives for the field. Given its value in scientific inquiry and technological application, this research is expected to attract widespread attention and citations from scientists in the field.
